# Rapidly Deploying a Neural Search Engine for the COVID-19 Open Research Dataset: Preliminary Thoughts and Lessons Learned

**Edwin Zhang,**[1] **Nikhil Gupta,**[1] **Rodrigo Nogueira,**[1] **Kyunghyun Cho,**[2,3,4,5] and **Jimmy Lin**[1]

[1] David R. Cheriton School of Computer Science, University of Waterloo
[2] Courant Institute of Mathematical Sciences, New York University
[3] Center for Data Science, New York University
[4] Facebook AI Research   [5] CIFAR Associate Fellow

## Abstract

We present the Neural Covidex, a search engine that exploits the latest neural ranking architectures to provide information access to the COVID-19 Open Research Dataset curated by the Allen Institute for AI. This web application exists as part of a suite of tools that we have developed over the past few weeks to help domain experts tackle the ongoing global pandemic. We hope that improved information access capabilities to the scientific literature can inform evidence-based decision making and insight generation. This paper describes our initial efforts and offers a few thoughts about lessons we have learned along the way.

## 1 Introduction

As a response to the worldwide COVID-19 pandemic, on March 13, 2020, the Allen Institute for AI released the COVID-19 Open Research Dataset (CORD-19) in partnership with a coalition of research groups.[1] With weekly updates since the initial release, the corpus currently contains over 47,000 scholarly articles, including over 36,000 with full text, about COVID-19 and coronavirus-related research more broadly (for example, SARS and MERS), drawn from a variety of sources including PubMed, a curated list of articles from the WHO, as well as preprints from bioRxiv and medRxiv. The stated goal of the effort is "to mobilize researchers to apply recent advances in natural language processing to generate new insights in support of the fight against this infectious disease". We responded to this call to arms.

In approximately two weeks, our team was able to build, deploy, and share with the research community a number of components that support information access to this corpus. We have also assembled these components into two end-to-end

---

[1] https://pages.semanticscholar.org/coronavirus-research

search applications that are available online at `covidex.ai`: a keyword-based search engine that supports faceted browsing and the Neural Covidex, a search engine that exploits the latest advances in deep learning and neural architectures for ranking. This paper describes our initial efforts.

We have several goals for this paper: First, we discuss our motivation and approach, articulating how, hopefully, better information access capabilities can contribute to the fight against this global pandemic. Second, we provide a technical description of what we have built. Previously, this information was scattered on different web pages, in tweets, and ephemeral discussions with colleagues over video conferences and email. Gathering all this information in one place is important for other researchers who wish to evaluate and build on our work. Finally, we reflect on our journey so far—discussing the evaluation of our system and offering some lessons learned that might inform future efforts in building technologies to aid in rapidly developing crises.

## 2 Motivation and Approach

Our team was assembled on March 21, 2020 over Slack, comprising members of two research groups from the University of Waterloo and New York University. This was a natural outgrowth of existing collaborations, and thus we had rapport from the very beginning. Prior to these discussions, we had known about the CORD-19 dataset, but had not yet undertaken any serious attempt to build a research project around it.

Motivating our efforts, we believed that information access capabilities (search, question answering, etc.)—broadly, the types of technologies that our team works on—could be applied to provide users with high-quality information from the scientific literature, to inform evidence-based decision making

and to support insight generation. Examples might include public health officials assessing the efficacy of population-level interventions, clinicians conducting meta-analyses to update care guidelines based on emerging clinical studies, virologist probing the genetic structure of COVID-19 in search of vaccines. We hope to contribute to these efforts by building better information access capabilities and packaging them into useful applications.

At the outset, we adopted a two-pronged strategy to build both end-to-end applications as well as modular, reusable components. The intended users of our systems are domain experts (e.g., clinicians and virologists) who would naturally demand responsive web applications with intuitive, easy-to-use interfaces. However, we also wished to build component technologies that could be shared with the research community, so that others can build on our efforts without "reinventing the wheel". To this end, we have released software artifacts (e.g., Java package in Maven Central, Python module on PyPI) that encapsulate some of our capabilities, complete with sample notebooks demonstrating their use. These notebooks support one-click replicability and provide a springboard for extensions.

## 3 Technical Description

Multi-stage search architectures represent the most common design for modern search engines, with work in academia dating back over a decade (Matveeva et al., 2006; Wang et al., 2011; Asadi and Lin, 2013). Known production deployments of this architecture include the Bing web search engine (Pedersen, 2010) as well as Alibaba's e-commerce search engine (Liu et al., 2017).

The idea behind multi-stage ranking is straightforward: instead of a monolithic ranker, ranking is decomposed into a series of stages. Typically, the pipeline begins with an initial retrieval stage, most often using "bag of words" queries against an inverted index. One or more subsequent stages reranks and refines the candidate set successively until the final results are presented to the user.

This multi-stage ranking design provides a nice organizing structure for our efforts—in particular, it provides a clean interface between basic keyword search and subsequent neural reranking components. This allowed us to make progress independently in a decoupled manner, but also presents natural integration points.

### 3.1 Modular and Reusable Keyword Search

In our design, initial retrieval is performed by the Anserini IR toolkit (Yang et al., 2017, 2018),[2] which we have been developing for several years and powers a number of our previous systems that incorporates various neural architectures (Yang et al., 2019; Yilmaz et al., 2019). Anserini represents an effort to better align real-world search applications with academic information retrieval research: under the covers, it builds on the popular and widely-deployed open-source Lucene search library, on top of which we provide a number of missing features for conducting research on modern IR test collections.

Anserini provides an abstraction for document collections, and comes with a variety of adaptors for different corpora and formats: web pages in WARC containers, XML documents in tarballs, JSON objects in text files, etc. Providing simple keyword search over CORD-19 required only writing an adaptor for the corpus that allows Anserini to ingest the documents. We were able to implement such an adaptor in a short amount of time.

However, one important issue that immediately arose with CORD-19 concerned the granularity of indexing, i.e., what should we consider a "document", as the "atomic unit" of indexing and retrieval? One complication stems from the fact that the corpus contains a mix of articles that vary widely in length, not only in terms of natural variations, but also because the full text is not available for some documents. It is well known in the IR literature, dating back several decades (e.g., Singhal et al. 1996), that length normalization plays an important role in retrieval effectiveness.

Here, however, the literature *does* provide some guidance: previous work (Lin, 2009) showed that paragraph-level indexing can be more effective than the two other obvious alternatives of (a) indexing only the title and abstract of articles and (b) indexing each full-text article as a single, individual document. Based on this previous work, in addition to the two above conditions (for comparison purposes), we built (c) a paragraph-level index as follows: each full text article is segmented into paragraphs (based on existing annotations), and for *each* paragraph, we create a "document" for indexing comprising the title, abstract, and that paragraph. Thus, a full-text article comprising $n$ paragraphs yields $n + 1$ separate "retrievable units"

---

[2]http://anserini.io/

in the index. To be consistent with standard IR parlance, we call each of these retrieval units a document, in a generic sense, despite their composite structure. An article for which we do not have the full text is represented by an individual document in this scheme. Note that while fielded search (dividing the text into separate fields and performing scoring separately for each field) can yield better results, for expediency we did not implement this. Following best practice, documents are ranked using the BM25 scoring function.

Based on "eyeballing the results" using sample information needs (manually formulated into keyword queries) from the Kaggle challenge associated with CORD-19,[3] results from the paragraph index did appear to be better (see Section 4 for more discussion). In particular, the full-text index, i.e., condition (b) above, overly favored long articles, which were often book chapters and other material of a pedagogical nature, less likely to be relevant in our context. The paragraph index often retrieves multiple paragraphs from the same article, but we consider this to be a useful feature, since duplicates of the same underlying article can provide additional signals for evidence combination by downstream components.

Since Anserini is built on top of Lucene, which is implemented in Java, our tools are designed to run on the Java Virtual Machine (JVM). However, TensorFlow (Abadi et al., 2016) and PyTorch (Paszke et al., 2019), the two most popular neural network toolkits, use Python as their main language. More broadly, Python—with its diverse and mature ecosystem—has emerged as the language of choice for most data scientists today. Anticipating this gap, our team had been working on Pyserini,[4] Python bindings for Anserini, since late 2019. Pyserini is released as a Python module on PyPI and easily installable via `pip`.[5]

Putting all the pieces together, by March 23, a scant two days after the formation of our team, we were able release modular and reusable baseline keyword search components for accessing the CORD-19 collection.[6] Specifically, we shared prebuilt Anserini indexes for CORD-19 and released

updated version of Anserini (the underlying IR toolkit, as a Maven artifact in the Maven Central Repository) as well as Pyserini (the Python interface, as a Python module on PyPI) that provided basic keyword search. Furthermore, these capabilities were demonstrated in online notebooks, so that other researchers can replicate our results and continue to build on them.

Finally, we demonstrated, also via a notebook, how basic keyword search can be seamlessly integrated with modern neural modeling techniques. On top of initial candidate documents retrieved from Pyserini, we implemented a simple *unsupervised* sentence highlighting technique to draw a reader's attention to the most pertinent passages in a document, using the pretrained BioBERT model (Lee et al., 2020) from the HuggingFace Transformer library (Wolf et al., 2019). We used BioBERT to convert sentences from the retrieved candidates and the query (which we treat as a sequence of keywords) into sets of hidden vectors.[7] We compute the cosine similarity between every combination of hidden states from the two sets, corresponding to a sentence and the query. We choose the top-$K$ words in the context, and then highlight the top sentences that contain those words. Despite its unsupervised nature, this approach appeared to accurately identify pertinent sentences based on context. Originally meant as a simple demonstration of how keyword search can be seamlessly integrated with neural network components, this notebook provided the basic approach for sentence highlighting that we would eventually deploy in the Neural Covidex (details below).

### 3.2 Keyword Search with Faceted Browsing

Python modules and notebooks are useful for fellow researchers, but it would be unreasonable to expect end users (for example, clinicians) to use them directly. Thus, we considered it a priority to deploy an end-to-end search application over CORD-19 with an easy-to-use interface.

Fortunately, our team had also been working on this, dating back to early 2019. In Clancy et al. (2019), we described integrating Anserini with Solr, so that we can use Anserini as a frontend to index directly into the Solr search platform. As Solr is also built on Lucene, such integration was not very onerous. On top of Solr, we were able to deploy

---

[3] https://www.kaggle.com/allen-institute-for-ai/CORD-19-research-challenge
[4] http://pyserini.io/
[5] https://pypi.org/project/pyserini/
[6] https://twitter.com/lintool/status/1241881933031841800

[7] We used the hidden activations from the penultimate layer immediately before the final softmax layer.

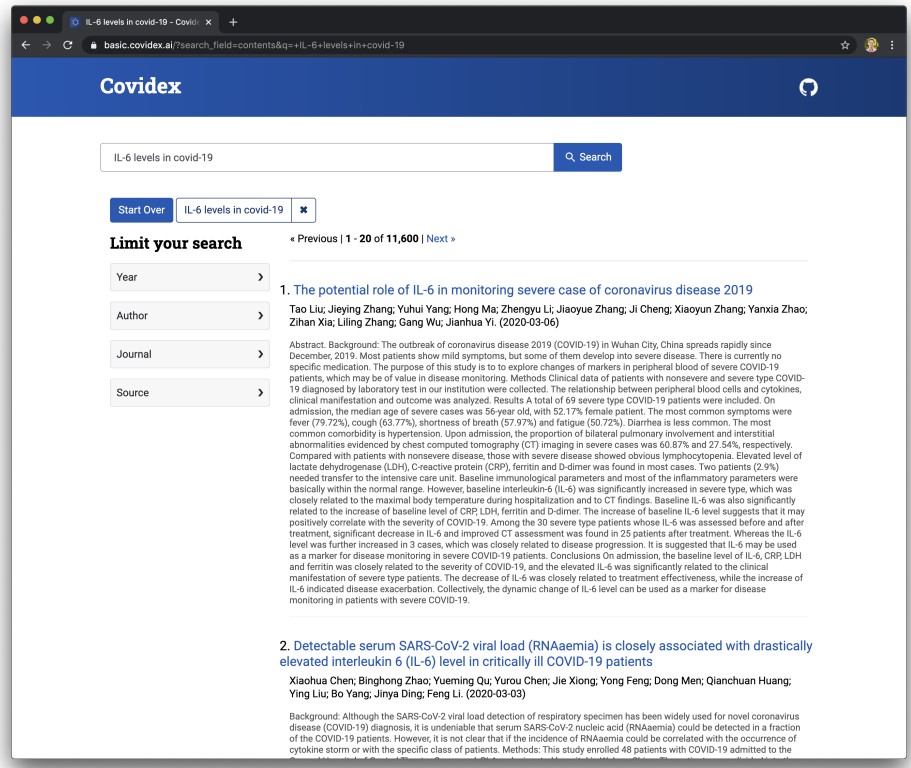

Figure 1: Screenshot of our "basic" Covidex keyword search application, which builds on Anserini, Solr, and Blacklight, providing basic BM25 ranking and faceting browsing.

the Blacklight search interface,[8] which is an application written in Ruby on Rails. In addition to providing basic support for query entry and results rendering, Blacklight also supports faceted browsing out of the box. With this combination—which had already been implemented for other corpora—our team was able to rapidly create a fully-featured search application on CORD-19, which we shared with the public on March 23 over social media.[9]

A screenshot of this interface is shown in Figure 1. Beyond standard "type in a query and get back a list of results" capabilities, it is worthwhile to highlight the faceted browsing feature. From CORD-19, we were able to easily expose facets corresponding to year, authors, journal, and source. Navigating by year, for example, would allow a user to focus on older coronavirus research (e.g., on SARS) or the latest research on COVID-19, and a combination of the journal and source facets would allow a user to differentiate between pre-prints and the peer-reviewed literature, and between venues with different reputations.

---

[8] https://projectblacklight.org/
[9] https://twitter.com/lintool/status/1242085391123066880

### 3.3 The Neural Covidex

The Neural Covidex is a search engine that takes advantage of the latest advances in neural ranking architectures, representing a culmination of our current efforts. Even before embarking on this project, our team had been active in exploring neural architectures for information access problems, particularly deep transformer models that have been pretrained on language modeling objectives: We were the first to apply BERT (Devlin et al., 2019) to the passage ranking problem. BERTserini (Yang et al., 2019) was among the first to apply deep transformer models to the retrieval-based question answering directly on large corpora. Birch (Yilmaz et al., 2019) represents the state of the art in document ranking (as of EMNLP 2019). All of these systems were built on Anserini.

In this project, however, we decided to incorporate our latest work based on ranking with sequence-to-sequence models (Nogueira et al., 2020). Our reranker, which consumes the candidate documents retrieved from CORD-19 by Pyserini using BM25 ranking, is based on the T5-base model (Raffel et al., 2019) that has been modified

to perform a ranking task. Given a query $q$ and a set of candidate documents $d \in D$, we construct the following input sequence to feed into T5-base:

$$\text{Query: } q \text{ Document: } d \text{ Relevant:} \qquad (1)$$

The model is fine-tuned to produce either "true" or "false" depending on whether the document is relevant or not to the query. That is, "true" and "false" are the ground truth predictions in the sequence-to-sequence task, what we call the "target words".

At inference time, to compute probabilities for each query–document pair (in a reranking setting), we apply a softmax only on the logits of the "true" and "false" tokens. We rerank the candidate documents according to the probabilities assigned to the "true" token. See Nogueira et al. (2020) for additional details about this logit normalization trick and the effects of different target words.

Since we do not have training data specific to CORD-19, we fine-tuned our model on the MS MARCO passage dataset (Nguyen et al., 2016), which comprises 8.8M passages obtained from the top 10 results retrieved by the Bing search engine (based on around 1M queries). The training set contains approximately 500k pairs of query and relevant documents, where each query has one relevant passage on average; non-relevant documents for training are also provided as part of the training data. Nogueira et al. (2020) and Yilmaz et al. (2019) had both previously demonstrated that models trained on MS MACRO can be directly applied to other document ranking tasks. We hoped that this is also the case for CORD-19.

We fine-tuned our T5-base model with a constant learning rate of $10^{-3}$ for 10k iterations with class-balanced batches of size 256. We used a maximum of 512 input tokens and one output token (i.e., either "true" or "false", as described above). In the MS MARCO passage dataset, none of the inputs required truncation when using this length limit. Training the model takes approximately 4 hours on a single Google TPU v3-8.

For the Neural Covidex, we used the paragraph index built by Anserini over CORD-19 (see Section 3.1). Since some of the documents are longer than the length restrictions of the model, it is not feasible to directly apply our method to the *entire* text at once. To address this issue, we first segment each document into spans by applying a sliding window of 10 sentences with a stride of 5. We then obtain a probability of relevance

for each span by performing inference on it independently. We select the highest probability among these spans as the relevance probability of the document. Note that with the paragraph index, keyword search might retrieve multiple paragraphs from the same underlying article; our technique essentially takes the highest-scoring span across all these retrieved results as the score for that article to produce a final ranking of *articles*. That is, in the final interface, we deduplicate paragraphs so that each article only appears once in the results.

A screenshot of the Neural Covidex is shown in Figure 2. By default, the abstract of each article is displayed, but the user can click to reveal the relevant paragraph from that article (for those with full text). The most salient sentence is highlighted, using exactly the technique described in Section 3.1 that we initially prototyped in a notebook.

Architecturally, the Neural Covidex is currently built as a monolith (with future plans to refactor into more modular microservices), where all incoming API requests are handled by a service that performs searching, reranking, and text highlighting. Search is performed with Pyserini (as discussed in Section 3.1), reranking with T5 (discussed above), and text highlighting with BioBERT (also discussed in Section 3.1). The system is built using the FastAPI Python web framework, which was chosen for speed and ease of use.[10] The frontend UI is built with React to support the use of modular, declarative JavaScript components,[11] taking advantage of its vast ecosystem.

The system is currently deployed across a small cluster of servers, each with two NVIDIA V100 GPUs, as our pipeline requires neural network inference at query time (T5 for reranking, BioBERT for highlighting). Each server runs the complete software stack in a simple replicated setup (no partitioning). On top of this, we leverage Cloudflare as a simple load balancer, which uses a round robin scheme to dispatch requests across the different servers.[12] The end-to-end latency for a typical query is around two seconds.

On April 2, 2020, a little more than a week after publicly releasing the basic keyword search interface and associated components, we launched the Neural Covidex on social media.[13]

---

[10] https://fastapi.tiangolo.com/
[11] https://reactjs.org/
[12] https://www.cloudflare.com/
[13] https://twitter.com/lintool/status/1245749445930688514

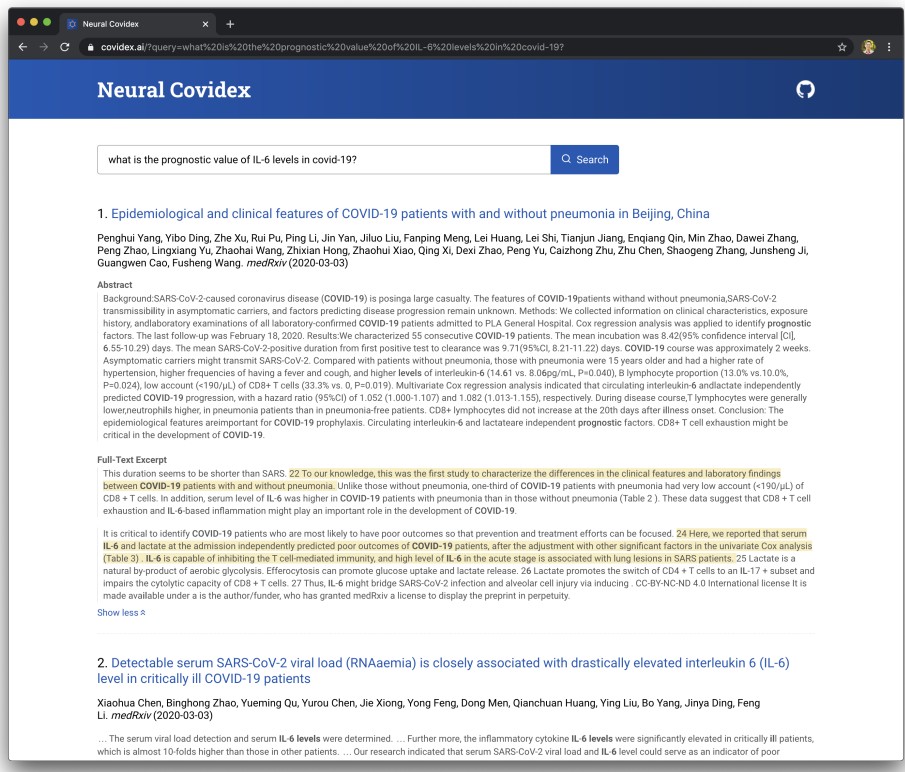

Figure 2: Screenshot of our Neural Covidex application, which builds on BM25 rankings from Pyserini, neural reranking using T5, and unsupervised sentence highlighting using BioBERT.

## 4 Evaluation or the Lack Thereof

It is, of course, expected that papers today have an evaluation section that attempts to empirically quantify the effectiveness of their proposed techniques and to support the claims to innovation made by the authors. Is our system any good? Quite honestly, we don't know.

At this point, all we can do is to point to previous work, in which nearly all the components that comprise our Neural Covidex have been evaluated separately, in their respective contexts (which of course is very different from the present application). While previous papers support our assertion that we are deploying state-of-the-art neural models, we currently have no conclusive evidence that they are effective for the CORD-19 corpus, previous results on cross-domain transfer notwithstanding (Yilmaz et al., 2019; Nogueira et al., 2020).

The evaluation problem, however, is far more complex than this. Since Neural Covidex is, at its core, a search engine, the impulse would be to evaluate it as such: using well-established methodologies based on test collections—comprising topics (information needs) and relevance judgments (human annotations). It is not clear if existing test collections—such as resources from the TREC Precision Medicine Track (Roberts et al., 2019) and other TREC evaluations dating even further back, or the BioASQ challenge (Tsatsaronis et al., 2015)—are useful for information needs against CORD-19. If no appropriate test collections exist, the logical chain of reasoning would compel the creation of one, and indeed, there are efforts underway to do exactly this.[14]

Such an approach—which will undoubtedly provide the community with valuable resources—presupposes that better ranking is needed. While improved ranking would always be welcomed, it is not clear that better ranking is the most urgent "missing ingredient" that will address the information access problem faced by stakeholders *today*. For example, in anecdotal feedback we've received, users remarked that they liked the highlighting that our interface provides to draw attention to the most salient passages. An evaluation of ranking, would not cover this presentational aspect of an end-to-end system.

---

[14] https://dmice.ohsu.edu/hersh/COVIDSearch.html

One important lesson from the information retrieval literature, dating back two decades,[15] is that batch retrieval evaluations (e.g., measuring mAP, nNDCG, etc.) often yield very different conclusions than end-to-end, human-in-the-loop evaluations (Hersh et al., 2000; Turpin and Hersh, 2001). As an example, a search engine that provides demonstrably inferior ranking might actually be quite useful from a task completion perspective because it provides other features and support user behaviors to compensate for any deficiencies (Lin and Smucker, 2008).

Even more broadly, it could very well be the case that search is completely the wrong capability to pursue. For example, it might be the case that users really want a filtering and notification service in which they "register" a standing query, and desire that a system "push" them relevant information as it becomes available (for example, in an email digest). Something along the lines of the recent TREC Microblog Tracks (Lin et al., 2015) might be a better model of the information needs. Such filtering and notification capabilities may even be more critical than user-initiated search in the present context due to the rapidly growing literature.

Our point is: we don't actually know how our systems (or any of its individual components) can concretely contribute to efforts to tackle the ongoing pandemic until we receive guidance from real users who are engage in those efforts. Of course, they're all on the frontlines and have no time to provide feedback. Therein lies the challenge: how to build improved fire-fighting capabilities for tomorrow without bothering those who are trying to fight the fires that already raging in front of us.

Now that we have a basic system in place, our efforts have shifted to broader engagement with potential stakeholders to solicit additional guidance, while trying to balance exactly the tradeoff discussed above. For our project, and for the community as a whole, we argue that informal "hallway usability testing" (virtually, of course) is still highly informative and insightful. Until we have a better sense of what users really need, discussions of performance in terms of nDCG, BLEU, and $F_1$ (pick your favorite metric) are premature. We believe the system we have deployed will assist us in understanding the true needs of those who are on the frontlines.

---

[15]Which means that students have likely not heard of this work and researchers might have likely forgotten it.

## 5   Lessons Learned

First and foremost, the rapid development and deployment of the Neural Covidex and all the associated software components is a testament to the power of open source, open science, and the maturity of the modern software ecosystem. For example, our project depends on Apache Lucene, Apache Solr, Project Blacklight, React, FastAPI, PyTorch, TensorFlow, the HuggingFace Transformers library, and more. These existing projects represent countless hours of effort by numerous individuals with very different skill sets, at all levels of the software stack. We are indebted to the contributors of all these software projects, without which our own systems could not have gotten off the ground so quickly.

In addition to software components, our efforts would not have been possible without the community culture of open data sharing—starting, of course, from CORD-19 itself. The Allen Institute for AI deserves tremendous credit for their tireless efforts in curating the articles, incrementally expanding the corpus, and continuously improve the data quality (data cleaning, as we all know, is 80% of data science). The rapid recent advances in neural architectures for NLP largely come from transformers that have been pretrained with language modeling objectives. Pretraining, of course, requires enormous amounts of hardware resources, and the fact that our community has developed an open culture where these models are freely shared has broadened and accelerated advances tremendously. We are beneficiaries of this sharing. Pretrained models then need to be fine-tuned for the actual downstream task, and for search-related tasks, the single biggest driver of recent progress has been Microsoft's release of the MS MARCO datatset (Nguyen et al., 2016). Without exaggeration, much of our recent work would not exist with this treasure trove.

Second, we learned from this experience that preparation matters, in the sense that an emphasis on good software engineering practices in our research groups (that long predate the present crisis) have paid off in enabling our team to rapidly retarget existing components to CORD-19. This is especially true of the "foundational" components at the bottom of our stack: Anserini has been in development for several years, with an emphasis on providing easily replicable and reusable keyword search capabilities. The Pyserini interface

to Anserini had also been in development since late 2019, providing a clean Python interface to Anserini. While the ability to rapidly explore new research ideas is important, investments in software engineering best practices are worthwhile and pay large dividends in the long run.

These practices go hand-in-hand with open-source release of software artifacts that allow others to replicate results reported in research papers. While open-sourcing research code has already emerged as a norm in our community, to us this is more than a "code dump". Refactoring research code into software artifacts that have at least some semblance of interface abstractions for reusability, writing good documentation to aid replication efforts, and other thankless tasks consume enormous amounts of effort—and without a faculty advisor's strong insistence, often never happens. Ultimately, we feel this is a matter of the "culture" of a research group—and cannot be instilled overnight—but our team's rapid progress illustrates that building such cultural norms is worthwhile.

Finally, these recent experiences have refreshed a lesson that we've already known, but needed reminding: there's a large gap between code for producing results in research papers and a real, live, deployed system. We illustrate with two examples:

Our reranking necessitates computationally-expensive neural network inference on GPUs at query time. If we were simply running experiments for a research paper, this would not be a concern, since evaluations could be conducted in batch, and we would not be concerned with how long inference took to generate the results. However, in a live system, both latency (where we test the patience of an individual user) and throughput (which dictates how many concurrent users we could serve) are critical. Even after the initial implementation of the Neural Covidex had been completed—and we had informally shared the system with colleagues—it required several more days of effort until we were reasonably confident that we could handle a public release, with potentially concurrent usage. During this time, we focused on issues such as hardware provisioning, load balancing, load testing, deploy processes, and other important operational concerns. Researchers simply wishing to write papers need not worry about any of these issues.

Furthermore, in a live system, presentational details become disproportionately important. In our initial deployment, rendered text contained artifacts of the underlying tokenization by the neural models; for example, "COVID-19" appeared as "COVID - 19" with added spaces. Also, we had minor issues with the highlighting service, in that sometimes the highlights did not align perfectly with the underlying sentences. These were no doubt relatively trivial matters of software engineering, but in initial informal evaluations, users kept mentioning these imperfections over and over again—to the extent, we suspect, that it was distracting them from considering the underlying quality of the ranking. Once again, these were issues that would have never cropped up if our end goal was to simply write research papers, not deploy a live system to serve users.

## 6  Conclusions

This paper describes our initial efforts in building the Neural Covidex, which incorporates the latest neural architectures to provide information access capabilities to AI2's CORD-19. We hope that our systems and components can prove useful in the fight against this global pandemic, and that the capabilities we've developed can be applied to analyzing the scientific literature more broadly.

## 7  Acknowledgments

This research was supported in part by the Canada First Research Excellence Fund, the Natural Sciences and Engineering Research Council (NSERC) of Canada, NVIDIA, and eBay. We'd like to thank Kyle Lo from AI2 for helpful discussions and Colin Raffel from Google for his assistance with T5.

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
