# OpenReview forum: "Rapidly Deploying a Neural Search Engine for the COVID-19 Open Research Dataset: Preliminary Thoughts and Lessons Learned"
_aclweb.org/ACL/2020/Workshop/NLP-COVID — NLP-COVID-2020 Abstractonly_

### Official Review · AnonReviewer1 · 2020-04-19
**Evaluation of a rapidly deployed search system for the CORD-19 data set**

**Rating:** 6
**Confidence:** 5

**Review:**

The authors have quickly stood up a conventional and neural search system based on the CORD-19 dataset, which is a commendable task given the global crisis caused by the Covid-19 pandemic. However, this paper is just a description of their system and design decisions, with some additional discussion of why a conventional TREC system-oriented retrieval evaluation would be limited. I agree with the authors on that point.

However, it would be helpful to have some sort of evaluation of their system to gauge whether their approach offers any novelty beyond the multiple other systems that been stood up and linked to from the Allen Institute site where the data is housed.

While I agree that a system-oriented evaluation approach would be limited, it would also be helpful to, say, compare the baseline and neural approaches. The first round of the TREC-COVID challenge evaluation is being conducted as I write this, and the first set of results will be available by late April or early May. I believe a better approach would be to include these results and then discuss their limitations.

If the authors do not believe the system-oriented results are important, they can explain why. They could provide some usage statistics for their system and describe other real-world use. They can also propose better evaluation studies, including those that involve users as they allude to in their paper.

Overall this is good work, but it could be much better, and we will hopefully learn more as the system is used, the test collection grows, and more complex tasks beyond ad hoc retrieval are evaluated with it.

---

> ### Author Response · Authors · 2020-04-22
> **Yup, we'll update with TREC-COVID results when they become available!**
>
> Dear Reviewer, thanks for your feedback.
>
> It is not the case that we believe that system-oriented results are not important, rather that, as you note and agree with, "a system-oriented evaluation approach would be limited" - for the reasons we have articulated in the current paper.
>
> Nevertheless, we are participating in TREC-COVID. We shall update the paper to incorporate results from round 1 when they become available, which is a bit less than two weeks from now.
>
> Best,
> Jimmy (on behalf of all the co-authors)

---

### Official Review · AnonReviewer3 · 2020-04-20
**Laudable effort, but currently more preprint material**

**Rating:** 5
**Confidence:** 5

**Review:**

This paper describes the rapid deployment of an IR system for the CORD-19 dataset. The system borrows state-of-the-art pieces, such as T5 and BioBERT, with training on MS MARCO. A search interface is shown that includes potential answer highlighting. Absent hard evaluation data, the authors provide more of a narrative of the system development, from initial conceptualization, early builds, deployment, and social media advertisement.

First and foremost, the authors are to be congratulated for this work. The effort is most laudable.

The style and tenor of the paper should be described as somewhere between a Medium post and a work-in-progress preprint. While normally this would be out of place, it is less out of place in a workshop such as this one. The casual language is fine and almost cute, while the frankness is quite welcome, but the other preprint-like or web post-like aspects of the paper are more problematic.

Let's start with the lack of evaluation, which the authors acknowledge that "It is, of course, expected that papers today have an evaluation section that attempts to empirically quantify the effectiveness of their proposed techniques". Of course, this empirical validation is what makes it science, as opposed to engineering/marketing. I certainly understand that the resource commitment to a TREC-style evaluation is beyond one group's ability under a pandemic situation. But the authors present the solution within their own paper: from footnote 14 it appears that the TREC-style evaluation is underway, and that should have preliminary results available well before this workshop's deadline. So why not simply issue this as a preprint to flagplant their admittedly laudable effort, then wait until some empirical results are available and submit *that* paper to the workshop?  If the authors plan to submit a work with results as a separate publication, this should be made more clear and a more compelling argument needs to be made for the scientific usefulness of the "behind the scenes" story of this search engine.

The authors make the claim that instead of the importance of search result ranking (despite Section 3 being about how great the system is at ranking on other tasks), that instead what is really important are the interface improvements that increase the usability of the system. However, every real-world IR evaluation has come to exactly this conclusion: experts always request various bells and whistles. First, note that users can't see the relevant articles that were missed, so they have little choice but to comment on the interface. But more importantly, good ranking != good usability, this is no novel claim. But usability evaluation is a well-honed science, and the authors do not perform any kind of usability evaluation either. These are less resource-intensive as well, typically requiring just an IRB protocol and some experts willing to provide feedback.

My main stylistic issue is that in many places this paper comes across as a brag-fest for the accomplishments of the group. These accomplishments are beyond doubt, of course, which is all the more reason why the authors can avoid tangents to cite unnecessary prior work (e.g., most of the BERT stuff in the first paragraph of Section 3.3, as this is not even used in the system).  The implications in Section 5 that the authors' group's "culture" and use of good software practices, unlike other poor-coding researchers, is also unnecessary (why is a "faculty advisor" reference even there?). One can extol the virtues of good culture and software engineering without coming across as bragging, so I would recommend a re-write of those parts of this paper to take a more scientific tone.

Minor:
- In Section 3.1 it is unclear what the "1" in "n+1" comes from. Do the authors also include a document that is just the title/abstract without the paragraphs?
- "who are engage in" -> "who are engaged in"

---

> ### Author Response · Authors · 2020-04-22
> **Yup, we'll update with TREC-COVID results when they become available!**
>
> Dear Reviewer, thanks for your feedback.
>
> As we've responded to reviewer 1 already, we are participating in TREC-COVID and will update the paper to incorporate results from round 1 when they become available.
>
> Best,
> Jimmy (on behalf of all the co-authors)

---

### Official Review · AnonReviewer2 · 2020-04-22
**Useful resources collected and introduced**

**Rating:** 6
**Confidence:** 5

**Review:**

This work is an introduction to a suite of resources and their corresponding ideas revolving around the COVID data provided by the Allen institute. Authors in particular provide Neural Covidex which allows exploration of information retrieval and NLP techniques over the COVID dataset.

The positives of this manuscript is as it also says gathering all the information and resources one would need to start exploring this dataset and further the techniques without having to spend weeks to put them all together. It is significant effort to get this far and of course authors are relying on their previous work over many years. There are some good pointers for some of the recent work in biomedical IR that is useful to know. I personally learnt from the content and enjoyed reading it.

This work however is written in haste (as it was needed due to the time limit of the current situation) and it is lacking the formal language of a scientific paper and of course evaluation. Obviously evaluation on this particular dataset is not straightforward given for example TREC is only starting on that effort. However though some initial analysis of the dataset could strengthen it. I was half expecting to see the discussion on evaluation metrics be a bit more mature and be given a separate section at least. This work as it stands, while very useful for the IR/NLP community it needs some reworking of the content to get to the state of publication in an ACL workshop.

---

> ### Author Response · Authors · 2020-04-22
> **Yup, we'll update with TREC-COVID results when they become available!**
>
> Dear Reviewer, thanks for your feedback.
>
> Your comments are very much in line with those of reviewers #1 and #2. We'll update the paper with TREC results as soon as they are available.
>
> Best,
> Jimmy (on behalf of all the co-authors)

---

### Comment · Program_Chairs · 2020-04-29
**Revise and Resubmit**

Based on the feedback from the reviewers, we would ask that you submit a revised version of the paper when you are able to update it with results from TREC-COVID.

Thank you for your submission!

---

> ### Author Response · Authors · 2020-05-11
> **Please accept/reject on the basis of the submitted work at the time of submission**
>
> Dear organizers,
>
> After further consideration and internal discussions, we respectfully
> decline to submit a revision and request that you make the
> accept/reject decision based on our manuscript as submitted.
>
> Our primary rationale for this is that updating the draft with results
> from TREC-COVID would make those results ineligible for inclusion in a
> submission to (for example) EMNLP without running afoul of duplicate
> publication rules and thus raising the ire of EMNLP reviewers.
>
> We stand by the quality of our work at the time it was submitted. All
> research papers represent point-in-time snapshots, and we feel that it
> is inequitable to subject papers to criticisms based on evidence that
> did not yet exist at the time of submission. Such a reviewing standard
> would not be consistent with broader ACL review policies [1].
>
> To that end, we will refrain from discussing results from TREC-COVID
> Round 1, which were only available several weeks after our submission.
>
> Best,
> Jimmy (on behalf of all the co-authors)
>
> [1] https://www.aclweb.org/adminwiki/index.php?title=ACL_Policies_for_Submission,_Review_and_Citation

---

> > ### Comment · Program_Chairs · 2020-05-12
> > **Thank you for your comments**
> >
> > Dear Jimmy and other authors,
> >
> > You will be given the option to proceed only with Abstract publication in the Workshop Proceedings after acceptance, rather than the full manuscript. This would then not preclude submission of the full paper to other venues. (NB: we will update the details on the website to make this clear.) Our objective in organizing this workshop is to allow people the opportunity to discuss their COVID-19 related work. Some will appreciate the publication in the ACL Anthology, others will prefer to try for something other than a Workshop publication. We are not looking to limit dissemination of this work.
> >
> > The reviewers have raised concerns about the paper beyond the lack of evaluation over TREC-COVID, in particular suggesting other types of assessment including a usability study or a comparison with other available systems (either in terms of functionality or in terms of analysis of actual retrieval results). Hence we stand by the Revision decision for the manuscript as it stands, in particular given that this is a long paper submission. We hope the feedback will be useful to you in preparing a submission to another venue.
> >
> > We will look into an "Abstract only" acceptance option, which we do not currently have.
> >
> > Best regards,
> > The Organizers

---

### Decision · Program_Chairs · 2020-05-13

**Decision:**

Accept (Abstract only)

**Comment:**

Per our discussion, we will proceed to an Abstract-only acceptance for this submission.

If you decide you wish to submit a revision of the paper here, please let us know.

---

> ### Author Response · Authors · 2020-05-13
> **Fair enough**
>
> This seems like a fair decision. Please convey the parameters of what
> exactly an "Abstract-only" final proceedings version means: Some
> version of the manuscript will still appear in the ACL Anthology, but
> with a drastically reduced page limit?

---

> > ### Comment · Program_Chairs · 2020-05-14
> > **Abstract-only, practically**
> >
> > Technically, an abstract-only inclusion would mean Title, Authors + Affiliation, Abstract text. (This is often used in biomedical conference proceedings, for instance)
> >
> > However in the context of ACL Proceedings I think it is reasonable to allow a full page, to allow room for references and a concise overview of the work. Does that suit you?
> >
> > Thanks.